# Risk Factors for the Development of Persistent Stuttering: What Every Pediatrician Should Know

**DOI:** 10.3390/ijerph19095225

**Published:** 2022-04-25

**Authors:** Julia Biancalana Costa, Ana Paula Ritto, Fabiola Juste, Fernanda Chiarion Sassi, Claudia Regina Furquim de Andrade

**Affiliations:** 1Division of Oral Myology—Hospital das Clinicas, School of Medicine, University of São Paulo, Sao Paulo 05403-010, Brazil; julia.biancalana@hc.fm.usp.br (J.B.C.); ana.ritto@fm.usp.br (A.P.R.); 2Department of Physiotherapy, Speech-Language and Hearing Sciece and Occupational Therapy School of Medicine, University of São Paulo, Sao Paulo 05403-010, Brazil; fjuste@usp.br (F.J.); fsassi@usp.br (F.C.S.)

**Keywords:** speech, speech fluency, stutter, persistence

## Abstract

Early identification and adequate treatment of children who stutter is important, since it has an impact on speech development. Considering the importance of aiding pediatricians to recognize children at risk for developing persistent stuttering, the aim of the present study was to correlate speech fluency characteristics of children, whose parents reported stuttering behaviors, to the risk factors of persistent stuttering. The participants were 419 children aged 2:0 to 11:11 years, who were divided into two groups: children with stuttering complaints (CSC), composed of children whose parents reported the presence of stuttering behaviors; and children with no stuttering complaint (CNSCs), composed of children with no stuttering behaviors. Risk variables were gathered based on a questionnaire answered by parents involving the following variables: sex, presence of family history of stuttering, whether stuttering behaviors were observed for more than 12 months, whether stuttering behaviors began before 5 years of age, increased effort to speak (i.e., syllable and sound repetitions and fixed articulatory positions), negative family attitude towards the child’s speech, and negative attitude towards the child’s own speech. The diagnosis of stuttering was determined by a formal speech assessment by a pathologist (SLP). The risk analysis indicated that increased effort to speak, negative family attitude towards the child’s speech, and complaints of stuttering for more than 12 months were associated with a higher risk of stuttering in children. Therefore, when pediatricians are faced with complaints about the presence of stuttering behaviors and these factors are present, they should immediately refer the patient to an SLP for specific assessment.

## 1. Introduction

Stuttering is defined as a neurodevelopmental disorder that affects the production of smooth speech flow [1]. The great majority of stuttering cases begins in early childhood, between 2 and 5 years old [2]. Of these children, 70–80% will recovery naturally from stuttering, without any treatment [3]. Most parents report the stuttering events to their children’s pediatrician before looking for a pathologist (SLP) [4,5].

Winters and Byrd [6] investigated pediatricians’ accuracy in identifying children who stutter and if these pediatricians would refer those children to a pathologist. The results showed that pediatricians would be more likely to refer to a pathologist the children who presented negative communication attitudes or who they heard stuttering. The study concluded that pediatricians should be educated about stuttering, including factors other than their own observation of stuttering behavior. 

Early identification and adequate treatment of children who stutter is important because it helps the development of positive attitudes toward communication and decreases the frequency of stuttering like-disfluencies (SLD) [4,7,8]. It is critical to identify children who present persistent stuttering, since the increase in SLD most certainly increases the risk of developing emotional, social, and academic problems [7,8,9,10,11]. 

Pediatrics is the medical specialty that takes care of the mental, physical, and social health of children. Pediatricians are responsible for preventive health care, diagnoses, and treatment of any acute and chronic diseases [12]. The American Academy of Pediatrics recommends that all professionals screen and monitor children in order to identify different developmental disorders and to refer the children to a specialist at a young age [13,14]. Considering the importance of aiding pediatricians to recognize children at risk for developing persistent stuttering, the aim of the present study was to correlate the speech fluency characteristics of children, whose parents reported stuttering behaviors, to the risk factors of persistent stuttering so that adequate referral to a pathologist can be made.

## 2. Materials and Methods

We conducted a prospective observational clinical study with children who stutter. This study received prior approval of the Institution’s Ethics Committee (CAPPesq—HCFMUSP Process no. 2.001.805) and informed consent was obtained from all participants or their families.

### 2.1. Participants

Participants were volunteers recruited from the pediatric and pathology clinic at the School of Medicine of the University of São Paulo. In order to be included in this study, individuals had to present the following characteristics: age between 2:0 and 11:11 (years–months), native speakers of Brazilian Portuguese, absence of any communication (language, speech, hearing) and of any neurological disorders, verified by a pathologist. 

Eligible participants for this study totaled 417 children. Participants were divided into two groups, as follows: children with stuttering complaints (CSC), composed of 217 children whose parents or caregivers reported a complaint about stuttering to the child’s pediatrician (parents observed that the child presents effort or strength to speak—i.e., sound repetitions, syllable repetitions, and fixed articulatory positions); children with no stuttering complaint (CNSC), composed of 200 children whose parents or caregivers did not have any stuttering complaints.

### 2.2. Procedures

#### 2.2.1. Stuttering Specific Variables

All of the participants’ parents and caregivers answered a few questions about the speech characteristics of the children. Based on the existing literature, questions involved specific stuttering variables such as: sex, presence of family history of stuttering (first- or second-degree relatives were considered positive), whether stuttering behavior had been observed for more than 12 months, if stuttering behavior began before 5 years of age, increased effort to speak (i.e., sound and syllable repetitions or fixed articulatory positions), negative family attitudes (negative meaning the family has a negative attitude, such as laughing at the child’s speech, interrupting the child, or saying “breathe, stay calm”), and negative child’s attitude (negative means the child stops speaking or is embarrassed about their speech). 

#### 2.2.2. Speech Material

In order to determine the diagnosis of stuttering, spontaneous speech samples were used for assessment and analysis. The samples for both groups were video-recorded using similar procedures. Individuals either produced a spontaneous monologue on a topic of their choice, or a sample was elicited by prompts given by a pathologist. Prior to the monologue, participants were given suggestions as to topics, such as family, toys, films, or hobbies. When necessary, the SLP asked questions to keep conversation going. With 2- and 3-year-old children, a toy was given to the children and a play situation with the SLP was recorded. The length of recordings was around 5 min (time necessary to obtain 200 fluent syllables). Orthographic transcriptions were carried out and the stuttering episodes within the 200 fluent syllables were marked on the transcripts. Single-word answers such as “yes” or “no” in response to prompting questions were excluded from analysis.

#### 2.2.3. Speech Sample Analysis

After orthographic transcriptions were carried out, each speech sample was analyzed considering the percentage of stuttered syllables. According to the existing literature, children who presented 3% or more of stuttered syllables were diagnosed as presenting stuttering. All speech samples were analyzed in a random order, regardless of the group to which they belonged, by two certified SLPs. The speech sample of 40 randomly selected children (i.e., 20 from each group) were re-evaluated using the Sander Agreement Index formula. Intrajudge reliability, for both the investigator and the one judge (a trained speech pathologist), ranged from 0.91 to 0.94. Interjudge reliability ranged from 0.90 to 0.95. All scores represent excellent levels of agreement.

### 2.3. Data Analysis

This study’s 417 participants were divided into children with stuttering behaviors (CSC—217 participants, 52.0% of the sample) and children with no stuttering behaviors (CNSC—200 participants, 48.0% of the sample). Data were analyzed in IBM SPSS software, version 27 (SSI Inc., Chicago, IL, USA). In addition to descriptive analysis, comparisons were performed between the groups, using Student’s *t*-test (for quantitative data) or Pearson’s chi-squared test (for categorical data). Significance was set at 5% for all analyses.

Secondarily, the possible risk factors were analyzed to identify which characteristics were the most significant predictors of stuttering (i.e., determined by the presence of 3% or more stuttered syllables at the SLP assessment, as described above) within the group of children with stuttering complaints. The backward stepwise logistic regression model was used to examine the relationships between independent variables (family complaints); the dependent variable was considered the presence of stuttering as diagnosed by the SLP assessment. Any variable having a significant univariate test at *p* ≤ 0.2 was selected as a candidate for the multivariate analysis. During the iterative multivariate fitting, covariates were removed from the model if they were non-significant at *p* ≤ 0.05 and not a confounder (i.e., did not change any remaining parameter estimates by more than 20%), using the backward stepwise selection method. The variables that remained in the model were considered independent risk factors.

## 3. Results

### 3.1. Comparison between Groups

Table 1 shows the results of the between-group analysis according to demographic variables. Significant between-group differences were found in the variable “sex”.

### 3.2. Risk Analysis

Within the group of children with stuttering behaviors (n = 217), 97 patients presented stuttering according to the SLP assessment (i.e., presented 3% or more stuttered syllables) and 120 did not present stuttering (i.e., presented less than 3% stuttered syllables at evaluation). Demographic and clinical data are presented in Table 2.

Table 3 presents the results of the multivariate logistic regression model for the prediction of stuttering. The univariate analysis identified three covariates initially as potential candidates for the multivariate model at the 0.2 alpha level based on the likelihood ratio statistic: sex, increased effort to speak, and negative family attitude. The table shows the initial results of the logistic regression model and the resulting model, containing only significant covariates. This analysis indicated that increased effort to speak and negative family attitude were associated with higher chances of presenting stuttering. Children with increased effort to speak and children with negative family attitudes are approximately 6.5 and 2.0 more likely to present stuttering, respectively.

## 4. Discussion

The aim of the current investigation was to correlate the speech fluency characteristics in children whose parents reported stuttering to the pediatrician with the risk factors of persistent stuttering, in order to aid pediatrician referral to a speech language pathologist. Overall, significant between-group differences were found in the variable “sex”. 

The risk analysis indicated that children with complaints of stuttering for more than 12 months, children with increased effort to speak, and children with negative family attitudes are 1.33, 1.08, and 1.29 more likely to present stuttering, respectively.

### 4.1. Comparison between Groups

Our results showed that the children with stuttering behaviors (CSC) featured more male children than the children with no stuttering behaviors (CNSC). Among studies on stuttering, it is well-known that male subjects are more likely to stutter than female. A recent review study showed that stuttering is 1.48 times more likely to persist in male than in female patients [15]. This is consistent with Chang’s findings about the differences between stuttering severity and white matter pathways for girls and boys who stutter [16]. Although our risk analysis did not find any results in the variable sex, the literature affirm that sex is an important risk variable for persistence stuttering and is correlated to differences in the maturation of speech motor control [17]. 

### 4.2. Risk Analysis

The results in the risk analysis indicated that increased effort to speak and negative family attitude were associated with higher chances of presenting the diagnosis of stuttering. According to the literature, the occurrence of disfluencies is common in children as a result of language development [18]. However, studies have reported that the presence of effort to speak at the ages of 4 and 5 years is an important risk indicator for persistent stuttering [17,19]. Singer [15] has also suggested that a higher amount of effort to speak indicates a greater concern for persistent stuttering.

The study found differences regarding family attitude. This is an important result, as it shows how important it is to advise parents. Negative family attitudes could lead to a negative attitude of the child towards their speech. A few studies have shown that negative parental attitude towards their child’s communication could also lead the child to present a negative attitude [15]. Our study suggests that the ‘negative family attitude’ is a risk variable for stuttering, which could impact children’s communication (frustration to speak or stop talking). It is the speech–language pathologists’ job to guide those parents and explain how speech and language development occurs and how they can help their child.

In this study, we did not find any differences in the variable ‘presence of family history of stuttering’. The results in the present study could have a bias. The children with no stuttering behaviors (CNSC) did not present any family history of stuttering. This could be a study limitation. According to the literature, stuttering is 1.89 times more likely to persist in children with a family history of stuttering than in children without any family history [15]. Therefore, it is an important predictor for persistence [2,17].

This study did not also find also any differences in the variables ‘complaint of stuttering for more than 12 months’ and ‘complaint of stuttering before 5 years of age’. That could be a possible bias because most of the children in this study were younger (mean age were 6.5 in both groups). If most of the participants are under 5 years old, this could be a study bias, and will not be presented as a risk factor for stuttering persistence. Future studies should cover different age groups. Although the present study did not find any results in the variable ‘complaint of stuttering before 5 years of age’, the age range of 4–5 years is an important period for changes in neurobehavioral system maturation. The literature shows that children with persistent stuttering usually present a later age of onset and stutter for more than 15 months [15,17]. Another study suggests that children with persistent stuttering will present a stable amount of effort to speak and children who recover will present a decline in the amount of effort to speak in the first 18 months after onset [3]. Pediatricians and pathologists should take that information into account when making decisions about whether treating the child or not [17].

A recent study by Singer [20] investigated the cumulative risk approach to predict whether a child who stutter will develop persistent stuttering. In the cumulative risk approach, the more predictive factors the child presents, the greater the chance that this child will develop persistence stuttering. The study concluded that there are four major predictive cumulative factors for this model to work: time since onset (less than 19 months), speech sound skills, expressive language skills, and stuttering severity (based on the Stuttering Severity Instrument—SSI-3). When two factors are presented in the child, this indicates a higher risk to develop persistent stuttering (93% sensitivity and 65% specificity). The more variables the child and family present, the greater the chance of the child being diagnosed with stuttering and needing specialized monitoring.

The pediatrician is the professional that interacts most with the child and their family. Most families will seek a pediatrician before seeking a speech pathologist. It is important to refer those children who present some of the characteristics mentioned above (male sex, stuttering for more than 12 months, increased effort to speak, and negative family attitude were associated with higher chances of presenting stuttering) to an SLP. Moreover, parents should be advised about the positive and negative attitudes that can help or cause damage to their child and how those attitudes could impact the way the child will relate with their own speech. We would like to consider that our study could help pediatricians to better refer families who complain about their children’s stuttering to an SLP and to minimize the negative impact of stuttering on children’s lives. 

Finally, our study had some limitations. First, the results of this study were derived from a single institution and may reflect some bias. Second, the speech analysis had evident limitations, since it was based exclusively on the first 200 fluent syllables of spontaneous speech samples. The assessment of other speaking situations, such as oral reading, single word naming, and longer speech samples, may produce different results. While these considerations should be noted, the ability to determine risk factors for the development of persistent stuttering and to aid pediatricians to decide when a referral to an SLP is necessary outweighs the current limitations.

## 5. Conclusions

The results of the present study indicate that when pediatricians are faced with complaints about the presence of stuttering behaviors, they should immediately refer the patient to an SLP for a specialized speech assessment when there is increased effort to speak, when negative family attitudes towards the child’s speech are present, and when stuttering behaviors have been observed for more than 12 months. 

## Figures and Tables

**Table 1 ijerph-19-05225-t001:** Between-group comparison according to demographic variables (N = 417).

		CSC (n = 217)	CNSC (n = 200)	*p*-Value
Age, in years—mean (±SD)		6.5 (±2.6)	6.5 (±2.9)	0.951
Sex—n (%)	Male	148 (68.2%)	88 (44.0%)	0.001 *
	Female	69 (31.8%)	112 (56.0%)	
Presence of Family History of Stuttering—n (%)		67 (30.9%)	0 (0.0%)	-

CSC: children with stuttering complaints; CNSC: children with no stuttering complaints; n: number of participants; SD: standard deviation. * Significant difference according to Pearson’s test.

**Table 2 ijerph-19-05225-t002:** Univariate comparison of children who did and did not present stuttering, within the group of children with stuttering behaviors (N = 217).

		3% or More Stuttered Syllables at Evaluation (n = 97)	Less than 3% Stuttered Syllables at Evaluation (n = 120)	*p*-Value
Age, in years—mean (±SD)		6.6 (±2.7)	6.4 (±2.6)	0.415
Sex—n (%)	Male	71 (73.2%)	77 (64.2%)	0.156
	Female	26 (26.8%)	43 (35.8%)	
Family Complaints—n (%)				
Presence of Family History of Stuttering		69 (71.1%)	81 (67.5%)	0.565
Behavior of stuttering for more than 12 months		81 (83.5%)	98 (81.7%)	0.723
Behavior of stuttering before 5 years of age		57 (58.8%)	66 (55.0%)	0.578
Increased effort to speak		93 (95.9%)	96 (80.0%)	<0.001 *
Negative family attitude		76 (78.4%)	82 (68.3%)	0.099
Negative child’s attitude		59 (60.8%)	76 (63.3%)	0.705

Note: n: number of participants; SD: standard deviation. * Significant difference according to Pearson’s test.

**Table 3 ijerph-19-05225-t003:** Multivariate logistic regression model for prediction of stuttering in children.

	First Iteration (Full Model)				2nd Iteration (Resulting Model)			
	OR	CI (95%)		*p*-Value	OR	CI (95%)		*p*-Value
		Lower	Upper			Lower	Upper	
Male Sex	1.434	0.779	2.638	0.247	-	-	-	-
Increased effort to speak	6.488	2.145	19.628	<0.001 *	6.516	2.156	19.691	<0.001 **
Negative family attitude	1.849	0.975	3.508	0.060	1.950	1.035	3.673	0.039 **

OR: odds ratio; CI: confidence interval. * Significant interaction according to multivariate logistic regression—full model. ** Significant interaction according to multivariate logistic regression—backward stepwise selection method.

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
