# Peer review of "Risk Factors for the Development of Persistent Stuttering: What Every Pediatrician Should Know"

_ijerph, 2022, doi:10.3390/ijerph19095225_

Round 1
Reviewer 1 Report
First of all, some editing of English language is required. Beginning by the same title. "Pedriatrians" is read with an "r" between "d" and "i". Other mistakes are "five years old8", "Table 2. shows", "can help or causa damages"... The format of the point "3.2 Comparison according to time since onset" is also out of style.
One outstanding problem of this article is that there is a reverse order of research as well as some repetition of other studies. Instead of stating the hipotheses, conducting the research, calculating the results, discussing them and obtaining the conclusions, it begins with significativity related to stuttering of gender, family attitude, span of the problem and age at which stuttering has begun. The research should be amplified to more causal variables and present the discussion of new cases.
Conclusions are simply tautological: the referral to pediatricians in case of male sex, stuttering for over a year, onset age 4 or 5 years, and negative family attitude. It perfectly links with what is read in the introduction "Results showed that pediatricinas would be more likely to refer to a speech-language pathologist the children who presented negative communication attitudes or who they heard stuttering" and "studies have shown factors that may contribute to persistence such as sex (it is most common in boys), family history of stuttering, later age of onset, longer time since onset, and type of stuttering-like disfluencies". Then, the research should add something to it.
Author Response
a) First of all, some editing of English language is required. Beginning by the same title. "Pedriatrians" is read with an "r" between "d" and "i". Other mistakes are "five years old8", "Table 2. shows", "can help or causa damages"... The format of the point "3.2 Comparison according to time since onset" is also out of style. – DONE – A review of the English language was performed.
b) One outstanding problem of this article is that there is a reverse order of research as well as some repetition of other studies. Instead of stating the hipotheses, conducting the research, calculating the results, discussing them and obtaining the conclusions, it begins with significativity related to stuttering of gender, family attitude, span of the problem and age at which stuttering has begun. The research should be amplified to more causal variables and present the discussion of new cases.
The manuscript was updated to include this suggestion. We performed a new statistical analysis to include the risk factors associated with the development of persistent stuttering. Based on this new analysis, the introduction, discussion and conclusion were updated. We also improved the methods section and removed information that involved specific details of the speech-language assessment.
2) Conclusions are simply tautological: the referral to pediatricians in case of male sex, stuttering for over a year, onset age 4 or 5 years, and negative family attitude. It perfectly links with what is read in the introduction "Results showed that pediatricinas would be more likely to refer to a speech-language pathologist the children who presented negative communication attitudes or who they heard stuttering" and "studies have shown factors that may contribute to persistence such as sex (it is most common in boys), family history of stuttering, later age of onset, longer time since onset, and type of stuttering-like disfluencies". Then, the research should add something to it.
The manuscript was updated to include this suggestion.
First, we would like to thank for the manuscript review. We agreed with the reviewer that the previews conclusions were simply tautological. So, in order to improve the manuscript we performed a new statistical analysis (risk analysis) to understand which characteristics the child whose parents have stuttering complaints have that are risk for persistence stuttering and the pediatrician should be more careful. We also corrected the English language and changed the title to ‘Risk factors for the development of persistent stuttering: what every pediatrician should know’. Finally, we changed the introduction and the discussion after the reviewer point out some errors and after the new statistics.
We strongly believe that the pediatrician may have education about the risk factors of persistence stuttering, in order to collaborate with medical referral and to improve children attitude towards theirs speech.
Reviewer 2 Report
Based on some analysis, this paper found differences in the variables sex and stuttering-like disfluencies. In the within-group analysis of the Research Group, this paper found differences in the vari-ables sex, age, and family attitude. When parents complain about their children’s stuttering, pedia-tricians should refer them to a speech pathologist in case of: male sex, stuttering for over a year, onset after age 4 or 5 years, and negative family attitude. Some comments are as follows:
[1] The semantics of Figure 1 are very simple and are not necessary
[2] The experimental results are simpler. If more influencing factors or experimental data are added, it will be more convincing
[3] If some excellent feature selection methods are used, they may have better results, refer to the following articles. This work should be mentioned at least in the conclusion and future work.
Xia S, Zhang Z, Li W, et al. GBNRS: A Novel Rough Set Algorithm for Fast Adaptive Attribute Reduction in Classification[J]. IEEE Transactions on Knowledge and Data Engineering, 2020.
Cao Y , Sun Y , Xie G , et al. A Sound-Based Fault Diagnosis Method for Railway Point Machines Based on Two-Stage Feature Selection Strategy and Ensemble Classifier[J]. IEEE Transactions on Intelligent Transportation Systems, 2021, PP(99):1-10.
[4] The shortcomings of this method should be analyzed in the conclusion, so as to facilitate readers' further research
Author Response
1. The semantics of Figure 1 are very simple and are not necessary - DONE
2.The experimental results are simpler. If more influencing factors or experimental data are added, it will be more convincing - The manuscript was updated to include this suggestion.
3. If some excellent feature selection methods are used, they may have better results, refer to the following articles. This work should be mentioned at least in the conclusion and future work. - The manuscript was updated to include this suggestion.
4. The shortcomings of this method should be analyzed in the conclusion, so as to facilitate readers' further research - DONE
First, we would like to thank for the manuscript review. In order to improve the manuscript we performed a new statistical analysis (risk analysis) to understand which characteristics the child whose parents have stuttering complaints have that are risk for persistence stuttering and the pediatrician should be more careful. We also corrected the English language, we took of the figure 1 and changed the title to ‘Risk factors for the development of persistent stuttering: what every pediatrician should know’. Finally, we changed the introduction and the discussion after the reviewer point out some errors and after the new statistics.
We strongly believe that the pediatrician may have education about the risk factors of persistence stuttering, in order to collaborate with medical referral and to improve children attitude towards theirs speech.
Round 2
Reviewer 2 Report
The author has made changes as requested, and it is recommended to accept.